# SINAI: SELECTIVE INJECTION OF NOISE FOR ADVERSARIAL ROBUSTNESS WITH IMPROVED EFFICIENCY

## ABSTRACT

Deep Neural Networks (DNNs) have revolutionized a wide range of industries, from healthcare and finance to automotive, by offering unparalleled capabilities in data analysis and decision-making. Despite their transforming impact, DNNs face two critical challenges: the vulnerability to adversarial attacks and the increasing computational costs associated with more complex and larger models. In this paper, we introduce an effective method designed to simultaneously enhance adversarial robustness and execution efficiency. Unlike prior studies that enhance robustness via uniformly injecting noise, we introduce a non-uniform noise injection algorithm, strategically applied at each DNN layer to disrupt adversarial perturbations introduced in attacks. By employing approximation techniques, our approach identifies and protects essential neurons while strategically introducing noise into non-essential neurons. Our experimental results demonstrate that our method successfully enhances both robustness and efficiency across several attack scenarios, model architectures, and datasets.

## 1 INTRODUCTION

Deep Neural Networks (DNNs) are at the forefront of technological advancements, powering a multitude of intelligent applications across various sectors (Al-Qizwini et al., 2017; Bai et al., 2018; Fujiyoshi et al., 2019). Yet, as DNNs become deeply integrated into mission-critical systems, two challenges emerge in DNN deployment. First, when DNNs are used in vital decision-making tasks, their vulnerability to adversarial attacks becomes a serious concern. From a self-driving car misinterpreting a traffic sign due to subtle, maliciously introduced perturbations, to defense systems being deceived into false detection, the outcomes could be catastrophic (Carlini & Wagner, 2017; Goodfellow et al., 2014; Kurakin et al., 2016; Madry et al., 2017; Moosavi-Dezfooli et al., 2016). Second, as DNNs become more complex and sophisticated, their computational demands increase correspondingly, calling for advanced optimization strategies to ensure that these powerful neural network models can operate efficiently even with constrained computational resources (Han et al., 2015; Niu et al., 2020; Roy et al., 2021; Zaheer et al., 2020).

Recent studies suggest that introducing noise not only enhances the robustness of DNN models but also provides a controllable means to achieve such improvement, as evidenced by recent research (Liu et al., 2018; He et al., 2019; Pinot et al., 2019; Xiao et al., 2020; Wu et al., 2020; Jeddi et al., 2020). Noise can be sampled from various probability distributions, including Gaussian (Liu et al., 2018; Lecuyer et al., 2019; He et al., 2019), Laplace (Lecuyer et al., 2019), Uniform (Xie et al., 2017), and Multinomial (Dhillon et al., 2018) distributions. Introducing random noise into the model, whether during training or inference phases, can be characterized as a non-deterministic querying process. Detailed theoretical proofs have shown that this approach leads to a significant improvement in adversarial robustness (Pinot et al., 2019).

However, these methods apply noise injections **uniformly** to all neurons, and such aggressive strategies inevitably compromise model accuracy (Pinot et al., 2019). Moreover, injecting noise into the model often requires additional computations (Liu et al., 2018; He et al., 2019; Jeddi et al., 2020), presenting a significant challenge to the model's efficiency, especially as the model size continues to grow.

The limitations raise a question: *Can we design a method to retain the benefits of noise injection while enhancing its execution efficiency?* To address this, we draw inspiration from techniques to

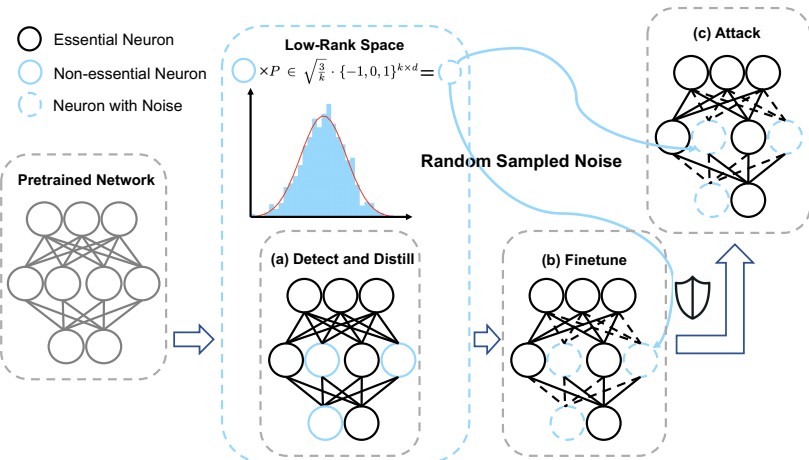

Figure 1: Overview of SINAI: Firstly, (a) we select non-essential neurons in the low-rank space and perform noise injection on these neurons. This step is efficiently integrated into the natural process of random projection, incurring minimal additional overhead. Secondly, (b) we return to the high-dimensional space for fine-tuning. Lastly, (c) the noises introduced during training and defense are randomized, which contributes to the improvement in robustness.

improve model execution efficiency and focus on the activation-based sparsity approach. This approach is based on the principle that not all neurons are equally important. Prior studies (Fu et al., 2021b; Guo et al., 2018; Madaan et al., 2020; Sehwag et al., 2020; Ye et al., 2019; Gopalakrishnan et al., 2018; Gui et al., 2019) have explored the potential of sparsity to enhance the robustness of DNNs. Yet, the application of sparsity to neural networks to boost adversarial robustness presents limitations and challenges in control. Moderate sparsity is essential for maintaining DNNs' adversarial robustness, and over-sparse networks become more vulnerable (Guo et al., 2018). We posit that this phenomenon is linked to the impact on essential neurons, which are crucial in preserving clean accuracy.

Thus, we hypothesize that only a subset of neurons are essential for representation learning while the rest can tolerate noise perturbations without affecting overall accuracy, which potentially bridges the benefits of noise injection and the concept of activation sparsity. To achieve this, we introduce **Injecting Noise to Non-Essential Neurons** to enhance DNNs robustness and efficiency. Instead of perturbing all neurons, our method protects identified essential neurons, bringing noise to only non-essential ones to enhance robustness.

The key is to identify the essential neurons and inject noise into the remaining non-critical neurons effectively and efficiently. As shown in Figure 1 (a), we adopt a learning-based approximate method (Liu et al., 2020) to identify essential neurons. Additionally, as a way to reduce computation costs, in Figure 1 (b), we propose to directly replace non-essential neurons with approximate values used for detection, and this approach will naturally bring in Pseudo-Gaussian noise with minimal computational resources. We further investigate the noise injection granularity and propose structured noise injection to significantly improve efficiency.

In summary, the contributions of this paper are as follows:

- We draw inspiration from the approaches of noise injection and sparsity and propose a novel method to simultaneously enhance both adversarial robustness and model efficiency, dubbed **S**elective **I**njection of **N**oise for **A**dversarial Robustness with **I**mproved Efficiency (SINAI). Upon identifying essential and non-essential neurons, we retain the essential neurons while efficiently introduce noise into the non-essential ones.

- We design a novel algorithm, that efficiently selects non-essential neurons, and a selective noise injection is brought via approximation to enhance DNN adversarial robustness while preserving clean accuracy. In addition, our method is a general approach which can be applied to any pre-trained network without retraining from scratch. We conduct the hard-

ware performance analysis of our algorithm, producing promising results demonstrating its potential for efficiency.

- We showcase that using a variety of DNN models across different datasets and exposed to different adversarial attacks, our algorithm consistently exhibits significantly higher robustness compared to the baselines. For example, on ResNet-18 with CIFAR-10, SINAI improves 14.74% robust accuracy under PGD[20] and 15.49% under AutoAttack, surpassing overfitting adversarial training with 82.6% BitOPs reduction in computational costs.

## 2 BACKGROUND AND RELATED WORK

**Adversarial Attacks & Defense with Noise.** Adversarial attacks, as discussed before (Croce & Hein, 2020), pose a significant threat to the deployment of machine learning (ML) models. Without protective measures, ML models can experience a significant drop in accuracy, often exceeding 20%, even under basic attacks. To defend against adversarial attacks, various methods have been proposed. Noise Injection (Liu et al., 2018; He et al., 2019; Pinot et al., 2019; Xiao et al., 2020; Wu et al., 2020; Jeddi et al., 2020; Lecuyer et al., 2019; Xie et al., 2017; Dhillon et al., 2018) is an effective method where models are trained by introducing random sampled noise to the original weights. Introducing random noise into the model can be characterized as a non-deterministic querying process, which acts as a shield, enhancing the model's robustness.

For example, RSE (Liu et al., 2018) proposes enhancing robustness by introducing a 'Noise' layer at the inception of the convolution block. This 'Noise' layer injects random noise into the activation values of the preceding layer, thereby fortifying the model's robustness. Similarly, PNI (He et al., 2019) adds a learnable Gaussian noise to each weight matrix/input/activation value, further enhancing robustness and probing for the granularity of noise injection. Based on PNI, Learn2Perturb (Jeddi et al., 2020) presents a regularizer for estimating distribution parameters and progressively enhances noise distributions. It is evident that noise injection-based approaches often introduce additional computational overhead, which poses challenges in the current era of increasingly larger models. A recent development, Random Projection Filters (RPF) (Dong & Xu, 2023) proposes a noise injection technique based on random projection. This method employs random projection to introduce noise into a randomized subset of filters. While the RPF approach bears similarities to ours, our method distinguishes itself by selecting neurons for noise injection based on their activation values. Moreover, we note that the RPF approach comes with some drawbacks, including the introduction of extra computational overhead, the requirement for training from scratch, and the inability to select the specific filters for noise injection precisely.

**Robustness and Efficiency.** Researchers explore several methods to improve the performance of deep neural networks (DNNs), including reducing their size and enhancing their resilience to attacks. In the domain of sparsity and robustness, a study (Guo et al., 2018) investigates how introducing sparsity in DNN architectures can bolster their resilience against adversarial attacks. A more comprehensive analysis (Ye et al., 2019) examines various techniques such as adversarial training, robust regularization, and model compression methods including pruning, quantization, and knowledge distillation to enhance adversarial robustness. Some studies, inspired by the lottery hypothesis (Frankle & Carbin, 2018) and adversarial training (Madry et al., 2017; Shafahi et al., 2019; Wong et al., 2020), combine pruning or sparse masking with adversarial training to obtain a sparse subnetwork with robustness (Sehwag et al., 2020; Madaan et al., 2020; Fu et al., 2021b). Nevertheless, these methods offer only modest enhancements in robustness and present challenges in precisely controlling and further improving the model's robustness (Guo et al., 2018).

## 3 APPROACH

Building on the hypothesis that only a subset of neurons are essential for model inference, we can introduce noise injection to the non-essential neurons as a defense against adversarial attacks. Our challenge is to identify precisely these essential neurons. Drawing from approximation studies (Achlioptas, 2001; Ailon & Chazelle, 2009; Vu, 2016), we propose a learning-based method to identify essential neurons and inject noise into the non-essential neurons. Our approach not only retains model accuracy but also enhances adversarial robustness.

## 3.1 Learning-based Approximation Method

| **Algorithm 1** Fine-Tuning Process |
|---|
| **Require:** Model parameters $W$, $b$; input batch $X = [x_1, \ldots, x_B]$; learning rate $\eta$; distilled parameters $\tilde{W}$, $\tilde{b}$; random projection matrix $P$. |
| **Ensure:** Updated model parameters $W$, $b$, $\tilde{W}^Q$, $\tilde{b}^Q$. |
| 1: **while** not converged **do** |
| 2:      $[z_1, \ldots, z_B] = WX + b$ |
| 3:      $[\tilde{z}_1, \ldots, \tilde{z}_B] = \tilde{W}^Q P X^Q + \tilde{b}^Q$ |
| 4:      Get $Loss_{original}$ using $z$ and true labels |
| 5:      Get $L_{MSE} = \frac{1}{B} \sum_{i=1}^{B} \|z_i - \tilde{z}_i\|_2^2$ |
| 6:      Combined loss $Loss = Loss_{original} + 0.1 \cdot L_{MSE}$ |
| 7:      Update $W, b, \tilde{W}^Q, \tilde{b}^Q$, |
| 8: **end while** |

| **Algorithm 2** Inference Process |
|---|
| **Require:** Original parameters $W$, $b$; quantized noisy parameters $\tilde{W}^Q$, $\tilde{b}^Q$; threshold $\theta_{th}$ to determine $m$; random projection matrix $P$; current input $x$. |
| **Ensure:** Final output $z$ |
| 1: $x^Q = Q(x)$ |
| 2: $\tilde{z} = \tilde{W}^Q P x^Q + \tilde{b}^Q$ |
| 3: Generating $m$ according to Section 3.2 |
| 4: **for** each $m_i \in m$ **do** |
| 5:      **if** $m_i == 1$ **then** |
| 6:          $z_i = \phi(W[i,:]x + b^i)$ |
| 7:      **else** |
| 8:          $z_i = \tilde{z}_i$ |
| 9:      **end if** |
| 10: **end for** |

As shown in Algorithm 1, we introduce layer-wise approximation as $\tilde{z} = \tilde{W} P x + \tilde{b}$, where $\tilde{W} \in \mathbb{R}^{n \times k}$ and $\tilde{b} \in \mathbb{R}^n$ are trainable parameters, $P \in \sqrt{\frac{3}{k}} \cdot \{-1, 0, 1\}^{k \times d}$ is a sparse random projection matrix. Note that the approximate vector $\tilde{z}$ has the same dimension with the original output vector $z$. Since the approximate data is only used for the selection of essential neurons and injection of noise, we incorporate quantization to decrease the bit-width of approximation parameters to further reduce computation costs. Specifically, we apply a one-time quantization step on $\tilde{W}$ as well as $\tilde{b}$ to INT4 fixed-point arithmetic as $\tilde{W}^Q$ and $\tilde{b}^Q$.

**Learning process of approximation parameters.** The trainable parameters, $\tilde{W}^Q$ and $\tilde{b}^Q$, are learned through minimizing the mean squared error (MSE) as the optimization target:

$$L_{MSE} = \frac{1}{B} \|z - \tilde{z}\|_2^2 = \frac{1}{B} \|(Wx + b) - (\tilde{W} P x + \tilde{b})\|_2^2 \tag{1}$$

where B is the mini-batch size. The random projection matrix $P$ is not trainable and stays constant after initialization.

## 3.2 Selection of Essential Neurons and Noise Injection

After we obtain optimized approximation parameters, we can use the approximate results $\tilde{z}$ to estimate the importance of individual neurons and select those with higher magnitude among $\tilde{z}$. We can select essential neurons by comparing the approximate results with a threshold, which is the smallest one in the Top-K values. A neuron is regarded as essential if its approximation from $\tilde{z}$ is larger than the threshold. Specifically, we can generate a binary mask $m \in \{0, 1\}^n$, work as a map of essential neurons to keep and non-essential neurons to inject noise, where $m_i$ equals 1 when the neurons are essential while it switches to 0 when the neurons are non-essential. In this way, we can estimate which neurons are essential without actual computations.

Once we identify the non-essential neurons, we can inject noise as a kind of perturbation. Instead of adding a noise term drawn from a normal or uniform distribution to the outputs, we propose to directly populate the non-essential neuron with approximate values drawn from $\tilde{z}$ that are computed in the selection step. Details can be found in Algorithm 2. Overall, the outputs of a DNN layer under our method can be formulated as

$$z^{'} = z \odot m + \tilde{z} \odot (1 - m), \tag{2}$$

where the $\odot$ denotes point-wise multiplication.

## 3.3 PRESERVATION OF CLEAN ACCURACY

With the aforementioned algorithm, we can allow the essential neurons to maintain clean accuracy, while the non-essential neurons are injected with noise, which has a boosting effect on the adversarial robustness. In this section, we will provide the theoretical support for our method.

We start with the proof that there is almost no clean accuracy drop by using our essential neurons selection algorithm. Our essential neurons selection algorithm is an algorithm that dynamically selects in a low-dimensional space. To prove that this algorithm has almost no loss in clean accuracy is to demonstrate that this transformation in low-dimensional space has almost no effect on the accuracy of matrix-matrix multiplication or matrix-vector multiplication. To simplify matters, we will focus on treating each operation within a sliding window of the convolution layer or the entirety of the fully connected (FC) layer, considering them as individual basic optimization problems for a single input sample. Each output activation $z_i$ is generated by the inner production:

$$z_i = \varphi\left(\langle x_i, W_j \rangle\right) \tag{3}$$

where $x_i$ is the $i$-th row in the matrix of input feature maps and for FC layer, there is only one $x$ vector. $W_j$ is the $j$-th column of the weight matrix $W$, and $\varphi(\cdot)$ is the activation function, here we omit the bias for simplicity. After defining Eq. 3 in this way, since matrix-matrix multiplication or matrix-vector multiplication consists of inner products, all we have to prove is that there exists a mapping of lower dimensional spaces that still gives a good approximation to inner products in higher dimensional spaces.

In dimensional transformations, according to the relation between inner product and the Euclidean distance, preserving inner-product is same as keeping the Euclidean distance between two points, as shown in the following lemma.

**Lemma 1.** (Johnson, 1984). Given $0 < \epsilon < 1$, a set of $N$ points in $\mathbb{R}^d$ (i.e., all $x_i$ and $W_j$ ), and a number of $k > O\left(\frac{\log(N)}{\epsilon^2}\right)$, there exists a linear map $f : \mathbb{R}^d \Rightarrow \mathbb{R}^k$ such that $(1-\epsilon)\|x_i - W_j\|^2 \leq \|f(x_i) - f(W_j)\|^2 \leq (1+\epsilon)\|x_i - W_j\|^2$.

For any given $x_i$ and $W_j$ pair, where $\epsilon$ is a hyper-parameter to control the approximation error, i.e., larger $\epsilon \Rightarrow$ larger error. This lemma is a dimension-reduction lemma, named Johnson-Lindenstrauss Lemma(JLL)(Johnson, 1984), which states that a collection of points within a high-dimensional space can be transformed into a lower-dimensional space, where the Euclidean distances between these points remain closely preserved.

Random projection (Achlioptas, 2001; Ailon & Chazelle, 2009; Vu, 2016) has found extensive use in constructing linear maps $f(\cdot)$. In particular, the original $d$-dimensional vector is projected to a $k$-dimensional space, where $k \ll d$, utilizing a random $k \times d$ matrix $\mathbf{P}$. Consequently, we can reduce the dimension of all $x_i$ and $W_j$ by applying this projection.

$$f(x_i) = \frac{1}{\sqrt{k}}\mathbf{P}x_i \in \mathbb{R}^k, f(W_j) = \frac{1}{\sqrt{k}}\mathbf{P}W_j \in \mathbb{R}^k \tag{4}$$

The random projection matrix $\mathbf{P}$ can be generated from Gaussian distribution (Ailon & Chazelle, 2009). In this paper, we adopt a simplified version, termed as sparse random projection (Achlioptas, 2001; Bingham & Mannila, 2001; Li et al., 2006) with $Pr\left(\mathbf{P}_{pq} = \sqrt{s}\right) = \frac{1}{2s}; \quad Pr\left(\mathbf{P}_{pq} = 0\right) = 1 - \frac{1}{s}; \quad Pr\left(\mathbf{P}_{pq} = -\sqrt{s}\right) = \frac{1}{2s}$ for all elements in $\mathbf{P}$. This $\mathbf{P}$ only has ternary values that can remove the multiplications during projection, and the remaining additions are very sparse. Therefore, the projection overhead is negligible compared to other high-precision multiplication operations. Here we set $s = 3$ with 67% sparsity in statistics.

When $\epsilon$ in Lemma 1. is sufficiently small, a corollary derived from Johnson-Lindenstrauss Lemma (JLL) yields the following norm preservation:

**Corollary 1.** (Instructors Sham Kakade, 2009) For $\mathbf{Y} \in \mathbb{R}^d$. If the entries in $\mathbf{P} \subset \mathbb{R}^{k \times d}$ are sampled independently from $N(0, 1)$. Then,

$$Pr\left[(1-\epsilon)\|\mathbf{Y}\|^2 \leq \|\frac{1}{\sqrt{k}}\mathbf{P}\mathbf{Y}\|^2 \leq (1+\epsilon)\|\mathbf{Y}\|^2\right] \geq 1 - O\left(\epsilon^2\right). \tag{5}$$

where $\mathbf{Y}$ could be any $x_i$ or $W_j$. This implies that the preservation of the vector norm is achievable with a high probability, which is governed by the parameter $\epsilon$. Given these basics, we can have the inner product preservation as:

**Theorem 1.** Given a set of $N$ points in $\mathbb{R}^d$ (i.e. all $x_i$ and $W_j$), and a number of $k > O\left(\frac{\log(N)}{\epsilon^2}\right)$, there exists random projection matrix $\mathbf{P}$ and a $\epsilon_0 \in (0,1)$, for $0 < \epsilon \leq \epsilon_0$ we have

$$Pr\left[\left|\left\langle \frac{1}{\sqrt{k}}\mathbf{P}x_i, \frac{1}{\sqrt{k}}\mathbf{P}W_j \right\rangle - \langle x_i, W_j \rangle\right| \leq \epsilon\right] \geq 1 - O\left(\epsilon^2\right). \tag{6}$$

for all $x_i$ and $W_j$, which indicates the low-dimensional inner product $\langle \frac{1}{\sqrt{k}}\mathbf{P}x_i, \frac{1}{\sqrt{k}}\mathbf{P}W_j \rangle$ can still approximate the original high-dimensional one $\langle x_i, W_j \rangle$ in Eq. 3 if the reduced dimension is sufficiently high. Therefore, it is possible to calculate Eq. 3 in a low-dimensional space for activation estimation and select the essential neurons. The detailed proof can be found in the Appendix A.

### 3.4 Improvement of Adversarial Robustness

Regarding proof of robustness improvement, Pinot et al. (2019) have demonstrated that injecting noise into a deep neural network can enhance the model's resilience against adversarial attacks. A deep neural network can be considered as a probabilistic mapping $M$, which maps the input $\mathcal{X}$ to $\mathcal{Z}$ via $M : \mathcal{X} \rightarrow P(\mathcal{Z})$. According to Pinot et al. (2019), the risk optimization term of the model is defined as:

$$\text{Risk}(M) := \mathbb{E}_{(x,z) \sim \mathcal{D}}\left[\mathbb{E}_{z' \sim M(x)}\left[\mathbb{1}\!\!\!/\,(z' \neq z)\right]\right] \tag{7}$$

In the adversarial attack scenario, the model risk optimization term becomes the:

$$\text{Risk}_\alpha(M) := \mathbb{E}_{(x,z) \sim \mathcal{D}}\left[\sup_{\|\tau\|_{\mathcal{X}} \leq \alpha} \mathbb{E}_{z' \sim M(x+\tau)}\left[\mathbb{1}\!\!\!/\,(z' \neq z)\right]\right] \tag{8}$$

where $\tau$ is the adversarial perturbation applied to the input sample, $\alpha$ is treated as the upper limit of perturbation. After obtaining these, Theorem 1 in Pinot et al. (2019) proved that noise sampled from the Exponential Family can ensure robustness. Finally, the robustness of the neural network with noise injection can be expressed by the following theorem:

**Theorem 2.** (Pinot et al., 2019) Let $M$ be the probabilistic mapping at hand. Let us suppose that $M$ is robust, then:

$$|\text{Risk}_\alpha(M) - \text{Risk}(M)| \leq 1 - e^{-\theta}\mathbb{E}_x\left[e^{-H(M(x))}\right] \tag{9}$$

where $H$ is the Shannon entropy $H(p) = -\sum_i p_i \log(p_i)$.

This theorem provides a means of controlling the accuracy degradation when under attack, with respect to both the robustness parameter $\theta$ and the entropy of the predictor. Intuitively, as the injection of noise increases, the output distribution tends towards a uniform distribution for any input. Consequently, as $\theta \rightarrow 0$ and the entropy $H(M(x)) \rightarrow \log(K)$, $K$ is the number of classes in the classification problem, both the risk and the adversarial risk tend towards $1/K$. Conversely, when no noise is introduced, the output distribution for any input resembles a Dirac distribution. In this scenario, if the prediction for an adversarial example differs from that of a regular one, $\theta \rightarrow \infty$ and $H(M(x)) \rightarrow 0$. Therefore, the design of noise needs to strike a balance between preserving accuracy and enhancing robustness against adversarial attacks, which proves our motivation.

### 3.5 Implications for Efficient Execution

As indicated in Eq. 2, the computed pattern is a mixture of precise computation and approximate computation in the form of non-essential neurons noise injection. For precise computation, only essential neurons need to be computed and non-essential neurons are from approximation. Hence, we can skip precise computations of non-essential neurons, leading to potential performance speedup and energy saving in the similar spirit of accelerated execution of activation sparsity. In addition, our approximate method incurs a small amount of low-precision operations.

**On noise injection granularity.** While selective noise injection can improve execution efficiency from computation skipping of essential neurons, the unconstrained and unstructured noise injection patterns would increase the hardware design complexity and execution overheads from irregular data access and low data reuse.

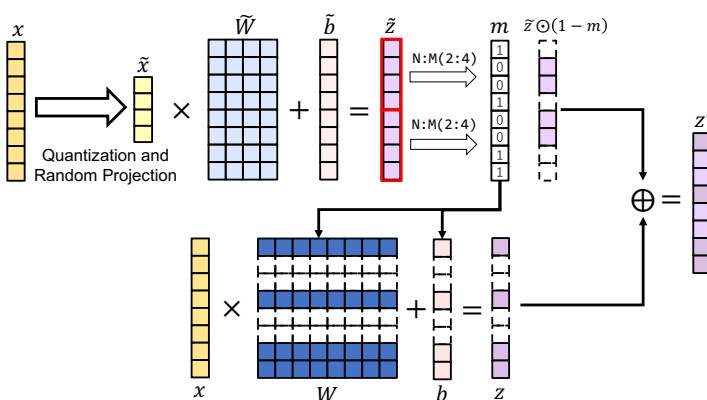

Figure 2: Structured Non-essential Neurons Noise Injection: After getting the approximate module from fine-tuning, the Top-K algorithm is utilized to take out the index of the largest value of $N$, and then the corresponding mask $m$ is generated. Accurate module carries out the N:M Sparsity through the mask $m$, and the final result is still a mixture of approximate and accurate modules.

Taking inspiration from the sparsity-oriented designs, we raise the hypothesis that noise injection granularity can be constrained in a similar way as sparsity constraints. In particular, structured sparsity, denoted as N:M, is an emerging trend that preserves N elements in every 1×M vector of a dense matrix. This fine-grained approach offers more combinations than block sparsity. An example is the 1:2 and 2:4 structured pruning techniques for neural network weight introduced in NVIDIA Ampere (Nvidia, 2020). Such techniques aim for efficiency and faster inference without sacrificing performance.

As shown in Figure 2, our revised method chooses N essential neurons out of a vector of M neurons and injects noise into the rest. Note that here we perform noise injection dynamically on neurons or activations, not static weight sparsity, despite the similarity in granularity.

## 4 EVALUATION

In this section, we evaluate the following two aspects of our selective noise injection method: firstly, to enhance adversarial robustness of the model while retaining clean accuracy; secondly, to demonstrate efficiency gains and implications on hardware of our method. In this paper, we focus our examination on ResNet-18 (He et al., 2016) and WideResNet34-10 (Zagoruyko & Komodakis, 2016), evaluating their performance on the CIFAR-10 and CIFAR-100 datasets (Krizhevsky et al., 2009) and ResNet-50 on ImageNet (Russakovsky et al., 2015). We keep the 10% essential neurons for experiments on CIFAR-10 and CIFAR-100, and 50% essential neurons on ImageNet to avoid clean accuracy loss. Our detailed evaluation methodology is in Appendix D. Additionally, we also design adaptive attack and examine our method for gradient obfuscation (Athalye et al., 2018a). For further details, refer to Appendix E (Adaptive Attack) and Appendix F (Gradient Obfuscation).

**Baselines.** For the comparison of adversarial robustness, we use the overfitting adversarial training (Rice et al., 2020) as a baseline. Moreover, our approach is mainly a kind of noise injection method, such that various noise injection methods are selected for comparison of adversarial robustness, including PNI (He et al., 2019), L2P (Jeddi et al., 2020). A recent method – Random Projection Filters (Dong & Xu, 2023) – is also a noise injection method. We reproduce all the above methods. For model efficiency comparisons, we evaluate the BitOPs for all the baselines and our method. Hardware experimental setting and more results could be found in Appendix C. As a defense method, we also compare our method with other state-of-the-art defense methods in recent years, such as TRADES (Zhang et al., 2019), FAT (Zhang et al., 2020), AWP (Wu et al., 2020) and RobustWRN (Huang et al., 2021).

### 4.1 RESULTS ON CIFAR-10/100

**Results on CIFAR-10.** As illustrated in Table 1 and Table 2, the underlined values represent the lowest robust accuracy for each method across all attackers, SINAI substantially improves robustness,

Table 1: The clean accuracy (%) and robust accuracy (%) of our non-essential neurons noise injection algorithm with ResNet-18 on CIFAR-10/100 under different adversarial attack methods. 'Improvement' is from the comparison of our method and overfitting adversarial training.

| Datasets | Methods | Clean | PGD$^{20}$ | FGSM | MIFGSM | CW$_{l2}$ | AutoAttack | Average | BitOPs |
|---|---|---|---|---|---|---|---|---|---|
| CIFAR-10 | OAT (Rice et al., 2020) | 81.71 | 52.53 | 57.08 | 55.54 | 78.27 | 48.62 | 58.41 | 2.60 E9 |
| | PNI (He et al., 2019) | 80.53 | 53.44 | 59.24 | 56.10 | 80.70 | 52.23 | 60.34 | 2.61 E9 |
| | L2P(Jeddi et al., 2020) | 80.82 | 53.59 | 61.42 | 56.52 | 81.40 | 65.47 | 63.68 | 2.61 E9 |
| | RPF (Dong & Xu, 2023) | 83.79 | 61.25 | 62.10 | 59.12 | 82.35 | 62.61 | 65.49 | 2.60 E9 |
| | SINAI (Ours) | 82.37 | 67.27 | 69.08 | 68.58 | 81.90 | 64.11 | **70.19** | **4.52 E8** |
| | *Improvement* | +0.66 | +14.74 | +12.00 | +13.04 | +3.63 | +15.49 | +11.78 | -82.6% |
| CIFAR-100 | OAT (Rice et al., 2020) | 54.85 | 28.92 | 31.31 | 30.28 | 50.48 | 24.65 | 33.13 | 2.60 E9 |
| | PNI (He et al., 2019) | 55.64 | 28.72 | 32.54 | 30.50 | 55.55 | 31.38 | 35.74 | 2.61 E9 |
| | L2P(Jeddi et al., 2020) | 55.00 | 28.50 | 34.01 | 30.34 | 55.03 | 38.51 | 37.28 | 2.61 E9 |
| | RPF (Dong & Xu, 2023) | 56.69 | 37.20 | 36.91 | 35.29 | 56.40 | 34.98 | 40.16 | 2.60 E9 |
| | SINAI (Ours) | 55.35 | 42.52 | 43.50 | 43.04 | 54.47 | 35.90 | **43.89** | **4.52 E8** |
| | *Improvement* | +0.50 | +13.60 | +12.19 | +12.76 | +3.99 | +11.25 | +10.76 | -82.6% |

Table 2: The clean accuracy (%) and robust accuracy (%) of our non-uniform noise injection algorithm with WideResNet-34-10 on CIFAR-10 under different adversarial attack methods. 'Improvement' is from the comparison of our method and overfitting adversarial training.

| Methods | Clean | PGD$^{20}$ | FGSM | MIFGSM | AutoAttack | Average | BitOPs |
|---|---|---|---|---|---|---|---|
| OAT (Rice et al., 2020) | 85.84 | 55.25 | 61.04 | 58.83 | 52.30 | 56.86 | 2.81 E10 |
| PNI (He et al., 2019) | 85.92 | 55.35 | 61.57 | 58.65 | 58.10 | 58.42 | 2.82 E10 |
| L2P(Jeddi et al., 2020) | 82.93 | 53.99 | 60.67 | 56.76 | 64.41 | 58.96 | 2.82 E10 |
| RPF (Dong & Xu, 2023) | 86.49 | 62.18 | 63.50 | 60.12 | 57.59 | 60.85 | 2.81 E10 |
| SINAI (Ours) | 86.56 | 74.95 | 76.02 | 75.61 | 69.26 | **73.96** | **4.88 E9** |
| *Improvement* | +0.72 | +19.7 | +14.98 | +16.78 | +16.96 | +17.10 | -82.6% |

and with an appropriately selected noise injection ratio, it maintains clean accuracy. Specifically, we can see that (1) our method consistently attains superior robust accuracy while maintaining comparable clean accuracy, largely outperforming the overfitting adversarial training method against prevalent white-box attacks across all networks and datasets. Moreover, compared to baseline noise injection approaches, our method demonstrates marked improvements against many attack methods. Specifically, on ResNet-18, our method not only maintains a clean accuracy of 82.37% but also significantly enhances robust accuracy compared to overfitting adversarial training. The improvements are 14.74% for PGD$^{20}$, 12.00% for FGSM, 13.04% for MIFGSM, 3.63% for CW$_{l2}$, and 15.49% for AutoAttack. Regarding noise injection baselines, our approach shows an improvement of 6.02% to 9.46% over the current state-of-the-art noise injection techniques under PGD$^{20}$, FGSM, and MIFGSM. Additionally, it achieves comparable robust accuracy when defending CW$_{l2}$ and AutoAttack. Similar to the results on ResNet-18, on WideResNet-34-10, our method also gains 12.77% and 1.2% improvement than Random Projection Filters under PGD$^{20}$ and AutoAttack respectively.

(2) Our method significantly reduces computational demands, thereby enhancing execution efficiency, e.g. 82.6% computational cost reduction. This indicates its capability to effectively adjust the degree of noise injection to attain the desired robustness, making it suitable for scenarios with limited computational resources.

**Results on CIFAR-100.** Table 1 shows the results of ResNet-18 on CIFAR-100, the observations on CIFAR-100 are consistent with those on CIFAR-10, demonstrating our method's scalability to more complex tasks. Our method improves 13.6% and 11.25% robust accuracy of overfitting adversarial training under PGD$^{20}$ and AutoAttack respectively without compromising clean accuracy. For other noise injection baselines, our approach also achieves significant improvements under the same attacks.

### 4.2 Scalability to Stronger Perturbations

We further evaluate our method's scalability to stronger perturbations with different PGD attacks $\epsilon$ and steps of ResNet-18 on CIFAR-10 and CIFAR-100. Figure 3 shows the results, it is worth emphasizing that noise injection on non-essential neurons achieves even higher robustness improvement. With a 90% noise injection ratio, our method achieves robustness improvements of 23.77% and 28.60% for two different attack $\epsilon = 12$ and $\epsilon = 16$, outperforming Overfitting Adversarial

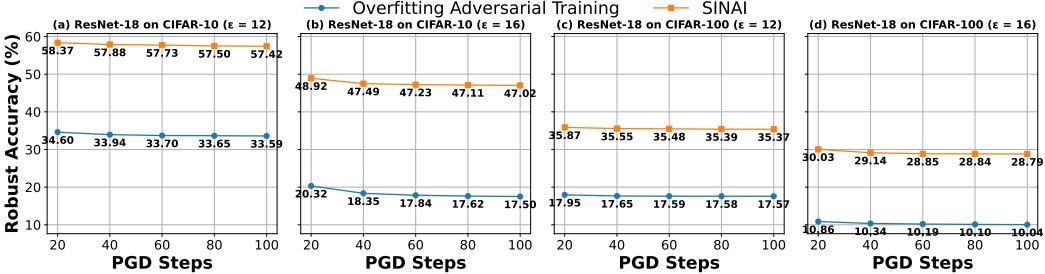

Figure 3: Evaluation stronger PGD attacks using different $\epsilon$ and steps. (ResNet-18 on CIFAR-10 and CIFAR-100)

Training. Similar trends are observed on CIFAR-100 (17.92% and 19.17%). This demonstrates the adaptability of our approach to defend stronger attack challenges.

## 4.3 COMPARE WITH SOTA DEFENSE METHODS

Compared with other SOTA defense methods, our approach demonstrates a more robust defense capability. Table 3 details the results between our method and other SOTA defense techniques applied to WideResNet-34-10 on CIFAR-10 and CIFAR-100, AutoAttack here keeps 10 steps. For instance, our method outperforms six baseline methods for WideResNet-34-10 on CIFAR-10 under AutoAttack, achieving improvements ranging from 1.2% to 16.97%. We further have the experiments of our method to ResNet-50 on ImageNet. For the baseline method, we reproduce Overfitting AT, Random Projection Filters, Double-win Quantization. For others we cite the number from the original paper. As shown in the Table 4, the noise injection ratio can be 50% without losing clean accuracy. Our method can get 6.05% and 6.18% improvement in the robust accuracy under PGD-10 and PGD-50, respectively. In addition, our method can achieve 39.5% saving in the BitOPs.

Table 3: Comparison with SOTA defense methods, except for overfitting AT and Random Projection Filters, all the baseline results are the reported ones in the original papers.

| Methods | CIFAR-10 | | CIFAR-100 | |
|---|---|---|---|---|
| | PGD[20] | AutoAttack | PGD[20] | AutoAttack |
| Overfitting Adversarial Training (Rice et al., 2020) | 55.25 | 52.52 | 31.27 | 27.79 |
| TRADES (Zhang et al., 2019) | 56.61 | 53.46 | - | - |
| FAT (Zhang et al., 2020) | 55.98 | 53.51 | - | - |
| AWP (Wu et al., 2020) | 58.14 | 54.04 | - | 28.86 |
| RobustWRN (Huang et al., 2021) | 59.13 | 52.48 | 34.12 | 28.63 |
| Random Projection Filters (Dong & Xu, 2023) | 62.18 | 68.25 | 31.53 | 40.35 |
| SINAI (Ours) | **74.95** | **69.45** | **48.02** | **42.28** |

Table 4: Comparison on ImageNet results.

| Methods | PGD-10 | PGD-50 | BitOPs |
|---|---|---|---|
| Overfitting Adversarial Training (Rice et al., 2020) | 40.68 | 39.83 | 3.34 E12 |
| RobustWRN (Huang et al., 2021) | 31.14 | - | 3.34 E12 |
| Double-Win Quantization(Fu et al., 2021a) | 43.05 | 42.97 | 1.73 E12 |
| Random Projection Filters(Dong & Xu, 2023) | 47.07 | 46.82 | 3.34 E12 |
| SINAI (Ours) | 46.73 | 46.01 | 2.02 E12 |
| Improvement | + 6.05 | + 6.18 | -39.5% |

## 4.4 ABLATION STUDY

In our algorithm, we quantize $\tilde{W}$ and $\tilde{b}$. To isolate the effect of this component on robustness, we first conduct a control experiment. Furthermore, the impact of noise injection in non-essential neurons, particularly the Noise Injection Ratio, affects both the clean and robust accuracy of the model. To investigate this effect in detail, we conduct ablation studies examining how variations in this ratio influence accuracy metrics, including robustness against PGD[20] attacks. Additionally, we consider the potential outcomes of applying noise injection to essential neurons and conduct further experiments to explore this aspect.

**Quantization and Robustness.** We test various quantization bit-width versions of $\tilde{W}$ and $\tilde{b}$, ranging from full precision to INT2. Our findings indicate that quantizing to INT4 retains the model's clean accuracy while offering the most significant efficiency improvements. As detailed in Table 5, although different quantization levels introduce varying levels of noise, they minimally impact the model's robust accuracy.

Table 5: ResNet-18 and WideResNet-34-10 on CIFAR-10 with different quantization

| Method | Base | FP32 (default) | INT32 | INT16 | INT8 | INT4 | INT2 |
|---|---|---|---|---|---|---|---|
| ResNet-18 on CIFAR-10 | | | | | | | |
| Clean Accuracy | 81.71 | 82.57 | 82.85 | 82.42 | 82.43 | 82.37 | 72.48 |
| PGD-20 | 52.53 | 67.28 | 67.36 | 67.15 | 67.03 | 67.27 | 54.86 |
| WideResNet-34-10 on CIFAR-10 | | | | | | | |
| Clean Accuracy | 85.84 | 86.01 | 86.32 | 86.90 | 86.51 | 86.56 | 78.88 |
| PGD-20 | 55.25 | 73.81 | 73.98 | 74.31 | 75.01 | 74.95 | 63.21 |

**Noise Injection Ratio.** As shown in Figure 4 (a) and (b), the relationship between the noise injection ratio and changes in clean/robust accuracy is not linear. Notably, clean accuracy begins to decrease significantly when the noise injection ratio reaches 90%, whereas robust accuracy shows substantial improvement at a 60% noise injection ratio. This result highlights the stability of our algorithm, which achieves a 'triple-win' scenario by maintaining high robustness and clean accuracy, even at high levels of execution efficiency (noise injection ratio).

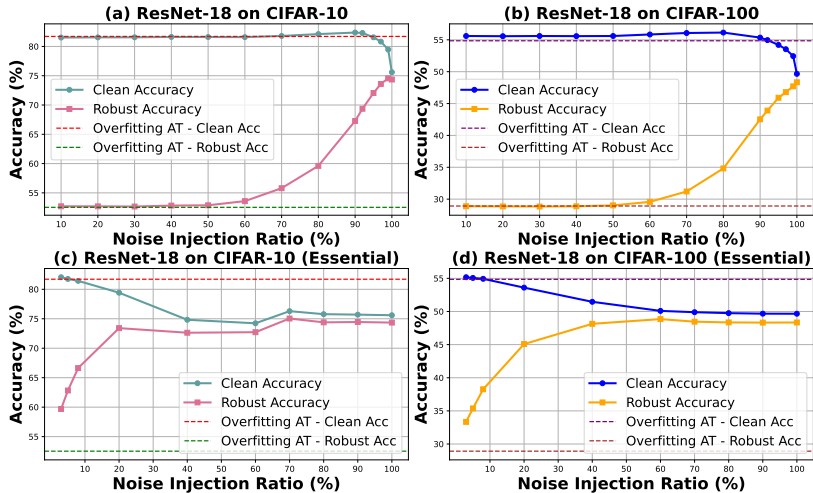

Figure 4: Clean and robust accuracy ($\text{PGD}^{20}$) under different noise injection ratio.

**Noise Injection to Essential Neurons.** Figure 4 (c) and (d) shows that the noise injection to essential neurons cannot maintain clean accuracy even with 20% noise injection ratio. However, we observe that PGD accuracy also increases with the proportion of noise injection. This likely occurs because injecting noise into essential neurons introduces drawbacks along the neural network's critical paths, which is worth further exploration in future studies.

## 5 CONCLUSION

In this work, we introduce a novel approach: noise injection on non-essential neurons, effectively bridging the gap between adversarial robustness and execution efficiency. Inspired by the approaches of noise injection and activation sparsity, our data-dependent method can precisely identify and keep the critical neurons that are contributing more to model accuracy, while injecting noise to the remaining trivial neurons with approximate values to improve robust accuracy. We believe that our findings highlight the importance of fine-grained noise injection, providing valuable insights into improving adversarial robustness and advancing the field of machine learning.

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

## A  PROOF OF ALGORITHM FOR INNER PRODUCT RESERVATION

**Theorem 1.** Given a set of $N$ points in $\mathbb{R}^d$ (i.e. all $x_i$ and $W_j$ ), and a number of $k > O\left(\frac{\log(N)}{\epsilon^2}\right)$, there exists random projection matrix $\mathbf{P}$ and a $\epsilon_0 \in (0,1)$, for $0 < \epsilon \leq \epsilon_0$ we have

$$Pr\left[\left|\left\langle \frac{1}{\sqrt{k}}\mathbf{P}x_i, \frac{1}{\sqrt{k}}\mathbf{P}W_j \right\rangle - \langle x_i, W_j\rangle\right| \leq \epsilon\right] \geq 1 - O\left(\epsilon^2\right).$$

for all $x_i$ and $W_j$.

**Proof.** According to the definition of inner product and vector norm, any two vectors $a$ and $b$ satisfy

$$\begin{cases} \langle \mathbf{a}, \mathbf{b}\rangle = \left(\|\mathbf{a}\|^2 + \|\mathbf{b}\|^2 - \|\mathbf{a} - \mathbf{b}\|^2\right)/2 \\ \langle \mathbf{a}, \mathbf{b}\rangle = \left(\|\mathbf{a} + \mathbf{b}\|^2 - \|\mathbf{a}\|^2 - \|\mathbf{b}\|^2\right)/2 \end{cases} . \tag{10}$$

It is easy to further get

$$\langle \mathbf{a}, \mathbf{b}\rangle = \left(\|\mathbf{a} + \mathbf{b}\|^2 - \|\mathbf{a} - \mathbf{b}\|^2\right)/4. \tag{11}$$

Therefore, we can transform the target in Eq. 3 to

$$|\langle f(x_i), f(W_j)\rangle - \langle x_i, W_j\rangle|$$

$$= \left|\|f(x_i) + f(W_j)\|^2 - \|f(x_i) - f(W_j)\|^2 - \|x_i + W_j\|^2 + \|x_i - W_j\|^2\right|/4 \tag{12}$$

$$\leq \left|\|f(x_i) + f(W_j)\|^2 - \|x_i + W_j\|^2\right|/4 + \left|\|f(x_i) - f(W_j)\|^2 - \|x_i - W_j\|^2\right|/4$$

which is also based on the fact that $|u - v| \leq |u| + |v|$. Now recall the definition of random projection in Eq. 4 of the main text

$$f(x_i) = \frac{1}{\sqrt{k}}\mathbf{P}x_i \in \mathbb{R}^k, \quad f(W_j) = \frac{1}{\sqrt{k}}\mathbf{P}W_j \in \mathbb{R}^k.$$

Substituting Eq. 4 into Eq. 12, we have

$$|\langle f(x_i), f(W_j)\rangle - \langle x_i, W_j\rangle|$$

$$\leq \left|\left\|\frac{1}{\sqrt{k}}\mathbf{P}x_i + \frac{1}{\sqrt{k}}\mathbf{P}W_j\right\|^2 - \|x_i + W_j\|^2\right|/4 + \left|\left\|\frac{1}{\sqrt{k}}\mathbf{P}x_i - \frac{1}{\sqrt{k}}\mathbf{P}W_j\right\|^2 - \|x_i - W_j\|^2\right|/4$$

$$= \left|\left\|\frac{1}{\sqrt{k}}\mathbf{P}(x_i + W_j)\right\|^2 - \|x_i + W_j\|^2\right|/4 + \left|\left\|\frac{1}{\sqrt{k}}\mathbf{P}(x_i - W_j)\right\|^2 - \|x_i - W_j\|^2\right|/4 \tag{13}$$

Further recalling the norm preservation in Eq. 5 of the main text: there exists a linear map $f : \mathbb{R}^d \Rightarrow \mathbb{R}^k$ and a $\epsilon_0 \in (0,1)$, for $0 < \epsilon \leq \epsilon_0$ we have

$$Pr\left[(1 - \epsilon)\|\mathbf{Y}\|^2 \leq \|\frac{1}{\sqrt{k}}\mathbf{P}\mathbf{Y}\|^2 \leq (1 + \epsilon)\|\mathbf{Y}\|^2\right] \geq 1 - O\left(\epsilon^2\right).$$

Substituting the Eq. 5 into Eq. 13 yields

$$P\left[\left|\left\|\frac{1}{\sqrt{k}}\mathbf{P}(x_i + W_j)\right\|^2 - \|x_i + W_j\|^2\right|/4 + \left|\left\|\frac{1}{\sqrt{k}}\mathbf{P}(x_i - W_j)\right\|^2 - \|x_i - W_j\|^2\right|/4 \dots\right.$$

$$\left. \leq \frac{\epsilon}{4}\left(\|x_i + W_j\|^2 + \|x_i - W_j\|^2\right) = \frac{\epsilon}{2}\left(\|x_i\|^2 + \|W_j\|^2\right)\right] \dots$$

$$\geq Pr\left(\left|\left\|\frac{1}{\sqrt{k}}\mathbf{P}(x_i + W_j)\right\|^2 - \|x_i + W_j\|^2\right|/4 \leq \frac{\epsilon}{4}\|x_i + W_j\|^2\right) \dots \tag{14}$$

$$\times Pr\left(\left|\left\|\frac{1}{\sqrt{k}}\mathbf{P}(x_i - W_j)\right\|^2 - \|x_i - W_j\|^2\right|/4 \leq \frac{\epsilon}{4}\|x_i - W_j\|^2\right) \dots$$

$$\geq \left[1 - O\left(\epsilon^2\right)\right] \cdot \left[1 - O\left(\epsilon^2\right)\right] = 1 - O\left(\epsilon^2\right).$$

Combining equation 12 and 14, finally we have

$$Pr\left[\left|\left\langle \frac{1}{\sqrt{k}}\mathbf{P}x_i, \frac{1}{\sqrt{k}}\mathbf{P}W_j \right\rangle - \langle x_i, W_j\rangle\right| \leq \frac{\epsilon}{2}\left(\|x_i\|^2 + \|W_j\|^2\right)\right] \geq 1 - O\left(\epsilon^2\right) \tag{15}$$

## B ROBUSTNESS EXPERIMENTS AT STRUCTURED GRANULARITY

In this section, in our pursuit of enhanced hardware execution efficiency, we experiment with structured design. This approach not only improves execution efficiency but also notably increases robustness compared with overfitting adversarial training. We assess our structured noise injection method with ratios ranging from 1:8 to 7:8, indicating noise injection into 1 to 7 out of every 8 neurons. Table 6 demonstrates that this approach of structured noise injection further enhances execution efficiency.

Table 6: The evaluation of clean accuracy, robustness and efficiency of different granularity of ResNet-18 on CIFAR-10 and CIFAR-100.

| Datasets | Method | Clean | PGD[20] | BitOPs |
|----------|--------|-------|---------|--------|
| CIFAR-10 | Noise Injection Ratio 10% | 81.52 | 52.71 | 2.53 E9 |
| | Noise Injection 1:8 | 81.55 | 52.78 | 2.47 E9 |
| | Noise Injection Ratio 20% | 81.55 | 52.70 | 2.27 E9 |
| | Noise Injection 2:8 | 81.54 | 52.94 | 2.14 E9 |
| | Noise Injection Ratio 30% | 81.56 | 52.67 | 2.01 E9 |
| | Noise Injection 4:8 | 81.53 | 55.22 | 1.49 E9 |
| | Noise Injection Ratio 50% | 81.61 | 52.88 | 1.49 E9 |
| | Noise Injection 6:8 | 82.11 | 58.07 | 8.42 E8 |
| | Noise Injection Ratio 80% | 82.11 | 59.61 | 7.12 E8 |
| | Noise Injection 7:8 | 82.18 | 65.73 | 5.17 E8 |
| | Noise Injection Ratio 90% | 82.37 | 67.27 | 4.52 E8 |
| CIFAR-100 | Noise Injection Ratio 10% | 55.60 | 28.88 | 2.53 E9 |
| | Noise Injection 1:8 | 55.62 | 28.83 | 2.47 E9 |
| | Noise Injection Ratio 20% | 55.57 | 28.86 | 2.27 E9 |
| | Noise Injection 2:8 | 55.62 | 28.97 | 2.14 E9 |
| | Noise Injection Ratio 30% | 55.60 | 28.84 | 2.01 E9 |
| | Noise Injection 4:8 | 55.80 | 29.58 | 1.49 E9 |
| | Noise Injection Ratio 50% | 55.60 | 29.02 | 1.49 E9 |
| | Noise Injection 6:8 | 56.07 | 33.20 | 8.42 E8 |
| | Noise Injection Ratio 80% | 56.14 | 34.82 | 7.12 E8 |
| | Noise Injection 7:8 | 55.51 | 39.56 | 5.17 E8 |
| | Noise Injection Ratio 90% | 55.35 | 42.52 | 4.52 E8 |

## C ADDITIONAL HARDWARE BACKGROUND

**Experimental Setup.** We tested the effectiveness of our algorithm on a co-designed hardware implementation via an in-house cycle-accurate simulator. The simulator tracks key performance metrics and maps them to specific power values. SRAM power/area is estimated using CACTI, and all other components are synthesized using Synopsys Design Compiler on the FreePDK 45nm PDK.

To get an accurate reference for an optimized accelerator, we design our own hardware. The general hierarchy is simple, with a central Matrix-Vector Unit (MVU), core-specific SRAM, and registers. The MVU is purpose-designed to match the algorithm requirements as efficiently as possible. It which uses 4-bit multipliers as a precision-scaling element, offering both 4- and 16-bit multiplication while limiting additional hardware.

**Hardware Efficiency.** By matching the hardware capabilities to the algorithm requirements, we can achieve better hardware efficiency. A major factor is using relatively lower cost of low-precision arithmetic vs. high-precision. Because multiplication characterizes each bit against each other, scaling from 4- to 16-bits increases the intensity by 16x, not 4x. As a result, the cost of approximation is low compared to full-precision calculation.

As it pertains to performance, Figure 5 shows that the noise injection granularity is directly tied to hardware performance. This figure highlights two key findings: 1) SINAI alone provides signifi-

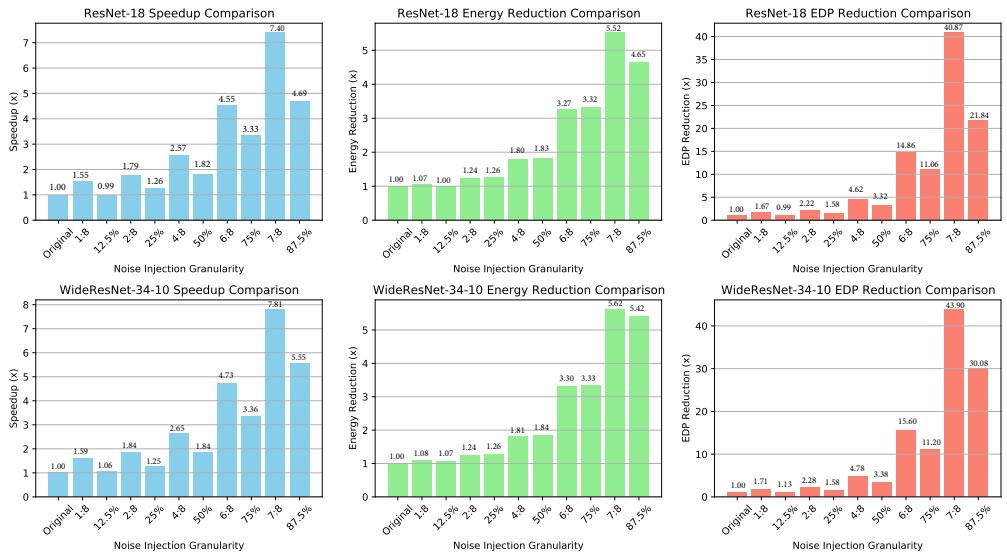

Figure 5: Normalized value of various metrics across different levels of perturbation generated from our hardware simulator. NOTE: sixteen BitOps = four 4-bit additions = one 4-bit multiplication.

cant performance gains and 2) applying structure at the same degree of noise further enhanced the efficiency.

First, with the relative intensity of full-precision calculation, increasing perturbation alone effectively improves speed and efficiency. This is because fewer precise values must be calculated. Because *high precision execution dominates performance*, a reduction in the high precision stage therefore has a significant effect on performance.

Second, when using a structured perturbation scheme, speed and efficiency are improved relative to an equivalent unstructured scheme. This is because with constrained, predictable execution patterns, we can expand the MVU without significant overhead, allowing for a higher throughput and faster execution despite a worse mapping efficiency. Additionally, with structured scheme, there is more data reuse, reducing the SRAM energy. Going further, with structured sparsity, we see that the enhanced parallelism allows for a greater reduction in execution time, similar energy, and higher efficiency between sparsities of the same degree.

From these two points, we can see structured dynamic perturbation is an effective means to vastly improve inference performance. This is because it capitalizes on the efficiency gains of sparse execution on high-cost operations while minimizing the energy & hardware costs which often limit the efficiency of sparsity.

A nuance to structured execution is that the optimal execution pattern is not always met. As seen in 5, structured does not always achieve as much of an EDP reduction in ResNet18 under high sparsity. This is due to the alignment and the reliance on minimum pipeline depth, which isn't consistently achieved under that sparsity. As such, VPUs will not be entirely full, leading to some wasted cycles. This can be avoided with some additional hardware, but that itself could outweigh the cost of occasionally reduced efficiency. As such, we perform our analysis with no such hardware. Regardless, we see that it does not significantly effect performance, as it accounts for a lower overhead relative to the savings.

## D    EVALUATION METHODOLOGY

**Networks and Datasets.** In this paper, we focus our examination on ResNet-18 (He et al., 2016) and WideResNet34-10 (Zagoruyko & Komodakis, 2016), evaluating their performance on the CIFAR-

10 and CIFAR-100 datasets (Krizhevsky et al., 2009) and ResNet-50 on ImageNet (Russakovsky et al., 2015).

**Training Strategy.** We adopt the state-of-the-art adversarial training protocol from overfitting adversarial training (Rice et al., 2020) for our experiments. Specifically, we train the network for 200 epochs using SGD with a batch size of 128, momentum of 0.9, a learning rate of 0.1, and weight decay set to $5 \times 10^{-4}$. The learning rate is decayed at 100 and 150 epochs with the decay factor 0.1. We also use PGD-10 for adversarial attack with a maximum perturbation size $\epsilon = 8/255$ and a step size of $2/255$. On ImageNet, we train the network for 90 epochs using SGD with a batch size of 64, momentum of 0.9, a learning rate of 0.02. We employ PGD-2 for adversarial example generation with a maximum perturbation size $\epsilon = 4/255$.

After obtaining the pre-trained model, SINAI is applied only to the convolutional layers, and we implemented these models using the PyTorch framework(Paszke et al., 2019). On CIFAR-10 and CIFAR-100, we distill the models using the momentum SGD optimizer for 50 epochs and fine-tuned them for only 1 epoch to achieve various noise injection ratios and structured sparsity. Our experiments with Non-Essential Neurons Noise Injection involve five different noise injection ratios: 90% and 99%. We also explore structured SINAI with patterns of 1:8 to 7:8, and the details can be found in Appendix B. On ImageNet, we distill our model using AdamW for 50 epochs and fine-tune it for 10 epochs.

**Attacks.** To evaluate adversarial robustness, we deploy various attack methods on our experiments, including Projected Gradient Descent (PGD) (Madry et al., 2017), Fast Gradient Sign Method (FGSM) (Goodfellow et al., 2014), C&W attack (Carlini & Wagner, 2017), Momentum-based Iterative Fast Gradient Sign Method (MIFGSM) (Dong et al., 2018), and AutoAttack (Croce & Hein, 2020). We use the torchattacks (Kim, 2020) library in Pytorch to deploy the above attack methods, with all parameters set according to the protocol given in the library. For FGSM, PGD, MIGFSM, and AutoAttack, we set the maximum perturbation size $\epsilon$ to $8/255$ and use a step size of $2/255$ for PGD and MIGFSM. PGD employs 20 steps, while MIGFSM uses 5 steps. For the CW attack, we set the learning rate to 0.01 and perform 1000 steps. For AutoAttack, we set the attack steps of APGD 20 steps. On ImageNet, $\epsilon$ is $4/255$ with steps 10 and 50.

# E  ADAPTIVE ATTACK

Our approach to crafting an adaptive attack is guided by the principle of 'T3:Adapt the objective to simplify the attack', as outlined in Tramer et al. (2020). This involves a thorough analysis of the defense mechanisms and design new objective functions, leading us to propose an adaptive attack specifically tailored to the injection of noise. To attack effectively, the attacker should be aware of the noise injected to the neurons. We estimate the noise impact on the output activations and define the final loss expression as:

$$\min_{\delta} \left( \mathcal{L}_C(x + \delta, y_t) - \mathcal{L}(A(Noisy(x + \delta)), A(x + \delta)) \right)$$

where:$\mathcal{L}_C$ is the classifier's loss, $Noisy$ is the function that applies noise to the neurons, $A$ is the activation.

The setting of adaptive PGD are the same as Vanilla PGD-20, epsilon = 8/255 and step size = 2/255. As illustrated in the table, the robust accuracy still achieves a more than 3.68% and 2.81% higher robust accuracy over Overfitting AT on CIFAR-10 and CIFAR-100, respectively, indicating the consistent effectiveness of our method.

Table 7: ResNet-18 on CIFAR-10 and CIFAR-100 under adaptive attack

| Method | CIFAR-10 (Clean) | CIFAR-10 (Adaptive PGD) | CIFAR-100 (Clean) | CIFAR-100 (Adaptive PGD) |
|---|---|---|---|---|
| Overfitting AT | 81.71 | 52.53 | 54.85 | 28.92 |
| SINAI | 82.37 | 56.21 | 55.35 | 31.73 |

# F  GRADIENT OBFUSCATION

We also evaluate our method on the gradient obfuscation.

1. EOT + PGD/AutoAttack experiments:

Applying Expectation over Transformation (EOT) (Athalye et al., 2018b) can correctly compute the gradient over the expected transformation to the input. EOT optimizes the expectation over the transformation $\mathbb{E}_{t\sim T} f(t(x))$. The optimization problem can be solved by gradient descent, noting that $\nabla \mathbb{E}_{t\sim T} f(t(x)) = \mathbb{E}_{t\sim T} \nabla f(t(x))$. A key insight here is that the gradient of the expectation can be expressed as the expectation of the gradient. This allows us to compute gradients not just through the classifier itself but also through the transformations applied to the input and approximating the expectation with samples at each gradient descent step. We set the EOT iteration 20 and try EOT+PGD and EOT+AutoAttack experiments for ResNet-18 on CIFAR-10 and CIFAR-100. We can see that the results are not much different.

Table 8: Performance on CIFAR-10 and CIFAR-100 under EOT+PGD

| Dataset | Method | AT | SINAI |
|---------|--------|-----|-------|
| CIFAR-10 | PGD | 52.53 | 67.27 |
| | EOT+PGD | 53.27 | 67.78 |
| CIFAR-100 | PGD | 28.92 | 42.52 |
| | EOT+PGD | 29.11 | 42.69 |

Table 9: Performance on CIFAR-10 and CIFAR-100 under EOT+AutoAttack

| Dataset | Method | AT | SINAI |
|---------|--------|-----|-------|
| CIFAR-10 | AutoAttack | 48.62 | 64.11 |
| | EOT+AutoAttack | 48.64 | 64.18 |
| CIFAR-100 | AutoAttack | 24.65 | 35.90 |
| | EOT+AutoAttack | 24.69 | 36.09 |

2. Checklist from Athalye et al. (2018a):

Table 10: Characteristics to identify gradient obfuscation

| Characteristics to identify gradient obfuscation | Pass | Fail |
|--------------------------------------------------|------|------|
| 1. One-step attack performs better than iterative attacks | ✓ | |
| 2. Black-box attacks are better than white-box attacks | ✓ | |
| 3. Unbounded attacks do not reach 100% success | ✓ | |
| 4. Random sampling finds adversarial examples | ✓ | |
| 5. Increasing distortion bound doesn't increase success | ✓ | |

For item 1, evidence from Tables 1 and 2 in our paper clearly shows that the FGSM attack (a one-step method) is less effective than the PGD attack (an iterative method).

Regarding item 2, we utilize the Square attack as a black-box method against ResNet on both CIFAR-10 and CIFAR-100 datasets. Our findings, as compared with Table 1, indicate that black-box attacks are generally less effective than white-box attacks.

Table 11: ResNet-18 on CIFAR-10 and CIFAR-100 under Square Attack

| Dataset | Method | AT | SINAI |
|---------|--------|-----|-------|
| CIFAR-10 | Square | 54.15 | 72.06 |
| CIFAR-100 | Square | 29.21 | 44.14 |

For item 3, we try unbounded attacks of PGD on our Noise Injection 90%, and the success rate result is 100%.

For item 4, the prerequisite is the gradient-based attack (e.g., PGD and FGSM) cannot find the adversarial examples, however the experiments in Fig. 3 reveals that our method still can be broken when increasing the distortion bound.

Lastly, item 5, also in Fig 3, increasing the distortion bound increases the attack success rate.

## G    LIMITATIONS AND FUTURE DIRECTIONS

We discuss the limitations of our approach and future directions.

**Apply to ViT architecture.** we have applied our method to the ViT architecture, while keeping our paper mostly on the exploration of CNN architectures. We design our application into two modules of transformer-based models. Firstly, on FFNs only, since the Attention layer in ViT plays the main role of pattern recognition, FFNs have little impact on the robustness of the whole network. Secondly, for applications involving the Attention layer, we hypothesize that the Softmax activation function mitigates the impact of noise through its averaging effect, further explaining the limited enhancement in robustness. This exploration serves as an early step, highlighting the necessity for further investigation into adapting our method for broader transformer architectures.

Table 12: Experiments of ViT-B-16 on CIFAR-10.

| Model | Clean | FGSM |
|---|---|---|
| Only FFN: | | |
| ViT-B-16 | 99.03 | 42.30 |
| Noise Injection 20% | 98.98 | 41.98 |
| Noise Injection 30% | 98.91 | 39.20 |
| Noise Injection 40% | 98.69 | 43.72 |
| Noise Injection 50% | 98.77 | 42.43 |
| Both | | |
| Noise Injection 20% | 98.25 | 42.57 |
| Noise Injection 30% | 98.00 | 44.37 |
| Noise Injection 40% | 97.66 | 40.92 |
| Noise Injection 50% | 96.57 | 38.23 |

