# OpenReview forum: "SINAI: Selective Injection of Noise for Adversarial Robustness with Improved Efficiency"
_ICLR.cc/2025/Conference — ICLR 2025 Conference Withdrawn Submission_

### Official Review · Reviewer_Sfxu · 2024-11-01

**Soundness:** 3
**Presentation:** 1
**Contribution:** 2
**Rating:** 3
**Confidence:** 4

**Summary:**

This work indicates that robustness can be enhanced by introducing noise into the internal layers of the target model. Additionally, it suggests that the weights can be divided into non-essential and essential neurons, which play different roles during model prediction.

**Strengths:**

This work provides extensive experiments to support the proposed idea.

**Weaknesses:**

1. **Poor Presentation:** In Section 1, the authors state, "we introduce injecting noise to non-essential neurons to enhance the robustness and efficiency of DNNs." However, they do not clearly explain why non-uniform noise is needed or the purpose of distinguishing between non-essential and essential neurons. The authors only provide a basic introduction to noise-based defense in Section 2.

2. **Unclear Motivation:** As mentioned previously, it is difficult for readers to follow why the proposed idea—an learning-based method to identify essential neurons and inject noise into non-essential neurons—is necessary. The AWP also introduces extra perturbations into each layer, which are non-uniform. The authors should clearly explain how their approach differs from prior works to strengthen the motivation.

3. **Unclear Implementation:** The paper introduces the proposed algorithm primarily with fully connected (FC) layers. However, in Wide Residual Networks (WRNs) or CNN-based networks, the FC layer typically appears only as the last layer. Does this imply that the proposed method can only be applied to the last layer? Additionally, the role of ϕ in Algorithm 2 is not defined in the main text.

4. **Unfair Comparison:** The attacking step for APGD is set to 20 in this work (lines 944-955), while the default setting is 100. This discrepancy suggests that the robustness may be overestimated.

5. **Overestimated Robustness:** If I understand correctly, random noise is injected in Algorithm 2 (step 2). However, AutoAttack is not a suitable method for assessing defenses that involve stochastic processes [*1].

[*1] https://github.com/fra31/auto-attack/issues/58

**Questions:**

A fair comparison is needed.

---

### Official Review · Reviewer_f6J5 · 2024-11-02

**Soundness:** 3
**Presentation:** 3
**Contribution:** 2
**Rating:** 5
**Confidence:** 3

**Summary:**

This paper introduces a method that enhances both adversarial robustness and computational efficiency in DNNs. The key idea is to selectively inject noise into non-essential neurons while preserving essential neurons, identified through a learning-based approximation method. The authors demonstrate improvements in robustness against various attacks while reducing computational costs.

**Strengths:**

1)	The paper proposes a new approach combining adversarial robustness and efficiency.
2)	The authors conduct extensive experiments across multiple datasets (CIFAR-10, CIFAR-100, ImageNet), and attack methods (PGD, FGSM, CW, MIFGSM, AutoAttack).

**Weaknesses:**

1)	While the technical contribution of this work is sound, a major weakness of the work is writing. Currently, the paper's writing suffers from two major issues: 1) The methodology is fragmented across multiple sections with poor transitions, making it difficult to understand the complete technical approach as a whole, and 2) The paper relies heavily on dense mathematical formulations without providing sufficient intuitive explanations, particularly in the theoretical sections 3.2 and 3.3.
2)	Does the proposed method improve execution time? Authors should report execution time in addition to the reported BitOPs.
3)	The paper doesn't explain how to optimally select the threshold for determining essential vs non-essential neurons
4)	The current evaluation on CNN is limited to ResNet based architectures. Some other CNN variants should be investigated

**Questions:**

Please refer to the weakness section.

---

### Official Review · Reviewer_YMx1 · 2024-11-03

**Soundness:** 3
**Presentation:** 2
**Contribution:** 2
**Rating:** 5
**Confidence:** 3

**Summary:**

This paper explores a new method to improve both the adversarial robustness and the execution efficiency of DNN by identifying essential and non-essential neurons. While essential neurons are protected, non-essential neurons are injected with noise. To identify essential neurons, the paper adopts the learning-based approximate method. In general, the proposed method can improve adversarial robustness and reduce computational costs.

**Strengths:**

- The motivation of the paper is clear and the idea behind why injecting noise into non-essential neurons intuitively makes sense.
- The paper is well-structured and easy to follow.
- Experiments were conducted against popular white-box attacks and the empirical improvements compared to baselines seem significant.

**Weaknesses:**

My main concerns are:
- Some of the theoretical parts do not support the authors' claims about the improvement in robustness and clearly help the clean accuracy preservation.
- The baseline is not SOTA and recent
- Some parts of the paper are not well written.

Please see the section on questions for a list of my detailed concerns.

**Questions:**

## Major
1. The contribution is not well written. The first and the second contributions are not distinguished enough. Both contributions state the proposed method is novel and retaining the essential neurons and employing selective noise injection to enhance both robustness and clean accuracy or model efficiency. The authors should rewrite and make the contribution more distinguished and concise. For example, you could propose that they focus on the novelty of the method in one point, and its specific benefits (robustness and efficiency) in another.
2. While baselines and competitors are adversarially trained models, the authors should clarify whether SINAI is applied to adversarially trained or non-adversarially trained models and provide results for both scenarios if possible. This would help validate your claim about the method's generalizability as the authors claimed the proposed method is general and can be applied to any pre-trained network without retraining from scratch. If the based model is non-adversarially trained, it seems the clean accuracy is very low e.g. the expected clean accuracy of resnet18 on CIFAR-10 should be higher 90%.
3. Some notations and terms are not well explained and described e.g. what is $Loss_{orignal}$ in algorithm 1 and $Q(x)$ in algorithm 2, and $k, d$ in line 182.  What is $P_{pq}$ in lines 256-257? What is $s$? Hence, I suggest the authors should include a notation table or provide explicit definitions when these terms are first introduced. This would significantly improve the paper's readability.
4. What is the convergence in algorithm 1? The paper lacks a complete definition of the condition of convergence.
5. In contribution, the authors state that the proposed method can be applied to any pre-trained model without retraining from scratch. Why are model parameters $W, b$ updated in algorithm 1?
6. How is sparse random projection $P$ randomly generated? Since theorem 1 confirms that a random projection matrix $P$ exists but not any $P$ such that the clean accuracy can be preserved, it is unclear if sparse random projection $P$ can guarantee the preservation of clean accuracy.
7. Section 3.4 and theorem 2 are general and useful but it is unclear how this can guarantee robustness improvement of SINAI since it does not state how SINAI improves adversarial robustness. The authors should clearly explain how theorem 2 can apply to SINAI. Since the noise needs to balance both clean accuracy preservation and adversarial robustness enhancement, the author should explain how the noise in SINAI helps obtain both objectives of improving clean accuracy and enhancing adversarial robustness?
8. The improvement in the evaluation section, particularly reported in Tables 1, 2 and 4, is misleading and not well explained. Since SINAI is not developed on top of overfitting adversarial training (OAT), why is the improvement compared with OAT? Additionally, OAT is fairly old and is not SOTA, the improvement of SINAI should be compared with SOTA methods such as RPF.
9. Some recent SOTA adversarially trained models [2, 3 and 4] have been demonstrated on Robustbench [1]. Since OAT is not SOTA, do the authors compare SINAI with one of the methods in [2, 3 and 4], or how does your method perform if one of them is incorporated into SINAI?
10. While the authors show the scalability of SINAI against two different perturbation budgets on CIFAR-10 and CIFAR-100 when compared with OAT, it is is unclear how it is compared with other SOTA methods e.g. RPF since RPF could achieve better robustness than SINAI at high perturbation budgets. Therefore, I suggest that the authors extend your comparisons on state-of-the-art methods like RPF, while still including OAT as a reference point. This would provide a clearer picture of SINAI's contributions to the current state of the field.
11. A threshold is used to identify essential and non-essential neurons but it is not well described and analysed in section 4 (Evaluation). How did the authors select m and threshold to obtain 10% essential neurons? Is it 10% neurons of each layer? It is unclear how well the proposed method performs with different percentages of essential neurons.
12. What is the noise injection ratio? How do you choose the best noise injection ratio for each dataset? The authors should provide an analysis of the noise injection ratio for Imagenet.

## Minor
1. If possible, please sort the citations in time order. For example, lines 077-079 or lines 123-124.
2. The experiment settings are not well described. For instance, it is unclear all attacks in Section 4 are l2 or linf attacks and what perturbation budgets are used in experiments in Section 4.1 and 4.3. Is the entire test set of each dataset (CIFAR-10, CIFAR-100 and ImageNet) used for evaluation? If not, the authors should describe it clearly.
3. In section 4.3, no clean accuracy is reported for Imagenet.

[1] https://robustbench.github.io/

[2] Bartoldson, Brian, James Diffenderfer, Konstantinos Parasyris and Bhavya Kailkhura. “Adversarial Robustness Limits via Scaling-Law and Human-Alignment Studies.” ICML2024

[3] Wang, Zekai, Tianyu Pang, Chao Du, Min Lin, Weiwei Liu and Shuicheng Yan. “Better Diffusion Models Further Improve Adversarial Training.” ICML2023

[4] Sehwag, Vikash, Saeed Mahloujifar, Tinashe Handina, Sihui Dai, Chong Xiang, Mung Chiang and Prateek Mittal. “Robust Learning Meets Generative Models: Can Proxy Distributions Improve Adversarial Robustness?” ICLR 2022.

---

### Official Review · Reviewer_yUfK · 2024-11-03

**Soundness:** 3
**Presentation:** 3
**Contribution:** 3
**Rating:** 6
**Confidence:** 4

**Summary:**

The authors propose a method for defending adversarial attacks by injecting random noise to "non-essential" neurons. They developed an algorithm to select "non-essential" neurons by training projecting to a low-dimensional space and training an approximation layer with quantization techniques applied. This method also has the benefit of reducing computational cost.

**Strengths:**

1. The presentation of this paper is good and easy to follow. The figures are intuitive and helpful. The algorithms are informative and compact.

2. The "non-essential neurons" is an interesting concept. And the results seem to imply that "essential" and "non-essential" neurons play different roles in model inference, where these "non-essential" neurons, although not contributing much to inference results on clean examples, can be exploited by adversaries. So that if we add noise only to "non-essential" neurons, we can achieve a better accuracy-robustness trade-off than injecting noise uniformly. If this is valid (which I think so based on session 4.4), I think it is an interesting and important finding revealing the key approach to achieve a better trade-off.

3. The experimental results are solid and rich. Especially I like ablation study 4.4 where I find answers to many of my previous question when reading the paper.

**Weaknesses:**

1. When talking about adversarial defense, it is critical to talk about the trade-off between clean accuracy and adversarial robustness. Especially for noise-injection type of approach, we know that different strength of noise-injection (e.g. ratio of injected neurons, size the random noise) will result in different clean accuracy and adversarial accuracy combinations. Therefore, instead of reporting a single point, it would be better to report the accuracy-robustness trade-off graphs by using different strength in noise injection. A single point many time cannot draw a solid conclusion. (please see details in the questions session)

2. It is not a weakness, but I wonder if there is a good way to "visualize" the different roles played by essential and non-essential neurons. E.g., would be great if we can show "non-essential neurons" become more active when processing adversarial examples than clean examples.

3. Again not a weakness but just a suggestion. It would be great to draw figure 4 as a acc-robustness trade-off graph where one axis is the clean acc and the other is the robust acc. It would more clear to demonstrate that injecting noise on "non-essential" neurons does achieves a better trade-off than injecting noise on essential neurons.

**Questions:**

1. What is the reason to use a low-dimension space to choose the "non-essential" neurons? Is it solely for computational cost consideration? What if we don't use an approximation layer but instead directly choose on the original layer?

2. On the accuracy-robustness trade-off issue mentioned in the "weakness" session.
2.1 On table 1, the clean acc. of SINAI and RPF are 82.37 and 83.79 while the acc. with PGD are 67.27 and 61.25. What if we inject less stronger noise to make the clean acc. of SINAI also around 83.79? Supposedly it should also lead to a weaker defense and lower acc. under PGD and it would be a more precise comparison to RPF.

2.2 Session 4.2 does not report clean acc. What is the clean acc. of methods compared in this session?

2.3 Similarly, session 4.3 does not report clean acc.

Would be happy to adjust my review scores if authors can address my questions.

---

### Note · Authors · 2024-11-19

**Comment:**

I have read and agree with the venue's withdrawal policy on behalf of myself and my co-authors.

**Withdrawal Confirmation:**

I have read and agree with the venue's withdrawal policy on behalf of myself and my co-authors.